# Structure, Properties, and Release Kinetics of the Polymer/Insect Repellent System Poly (l-Lactic Acid)/Ethyl Butylacetylaminopropionate (PLLA/IR3535)

**DOI:** 10.3390/pharmaceutics14112381

**Published:** 2022-11-04

**Authors:** Fanfan Du, Rafael Erdmann, Albrecht Petzold, Andre Wutzler, Andreas Leuteritz, Michael Nase, René Androsch

**Affiliations:** 1Interdisciplinary Center for Transfer-Oriented Research in Natural Sciences (IWE TFN), Martin Luther University Halle-Wittenberg, D-06099 Halle (Saale), Germany; 2Institute for Biopolymers and Sustainability (ibp), University of Applied Sciences Hof, D-95028 Hof (Saale), Germany; 3Institute of Physics, Martin Luther University Halle-Wittenberg, D-06099 Halle (Saale), Germany; 4Polymer Service GmbH Merseburg (PSM), D-06217 Merseburg, Germany; 5Leibniz-Institut für Polymerforschung Dresden e. V., Hohe Str. 6, D-01069 Dresden, Germany

**Keywords:** poly (l-lactic acid), ethyl butylacetylaminopropionate (IR3535), plasticization, mechanical properties, repellent release

## Abstract

The insect repellent ethyl butylacetylaminopropionate (IR3535) was used as a functional additive for poly (l-lactic acid) (PLLA) to modify its structure and mechanical properties and achieve insect repellency. PLLA/IR3535 mixtures at various compositions were prepared via melt extrusion. In the analyzed composition range of 0 to 23 m% IR3535, PLLA and IR3535 were miscible at the length scale represented by the glass transition temperature. Addition of IR3535 resulted in a significant decrease in the glass transition temperature of PLLA, as well as in the elastic modulus, indicating its efficiency as a plasticizer. All mixtures were amorphous after extrusion, though PLLA/IR3535 extrudates with an IR3535 content between 18 and 23 m% crystallized during long-term storage at ambient temperature, due to their low glass transition temperature. Quantification of the release of IR3535 into the environment by thermogravimetric analysis at different temperatures between 50 and 100 °C allowed the estimation of the evaporation rate at lower temperatures, suggesting an extremely low release rate with a time constant of the order of magnitude of 1–2 years at body temperature.

## 1. Introduction

The advances in the development of polymer materials over the last century have yielded many biomedical applications [1,2], among others, serving as platforms for advanced medical devices/drug delivery systems, allowing the controlled release of active compounds [3,4]. From a long-term environmental sustainability perspective, the use of biopolymers is the preferred alternative to petroleum-based materials [5,6]; moreover, biopolymers are expected to degrade in a reasonable time at the end of their lifecycle, avoiding adverse effects on the environment [7,8]. Poly (l-lactic acid) (PLLA)—an aliphatic thermoplastic polyester—is such a biopolymer, being widely used in different industrial fields, such as food packaging, agriculture, automotive, electronics, and biomedicine, including tissue engineering and drug delivery applications, because of its biodegradability and biocompatibility [9,10,11,12]. Despite these merits, PLLA has a few weaknesses, including brittleness and low crystallization rate, which limit its wider industrial application [10,11,12,13]. One of the most effective ways to increase the toughness, ductility, and processability of PLLA is by modifying the polymer with plasticizers [14,15]. Plasticizers require good miscibility with the polymer, low volatility, and adequate stability [16]. An efficient plasticizer is expected to reduce the glass transition temperature (*T*_g_) of the amorphous domains, such that at the temperature of application the polymer is in a rubbery state. Several substances have been studied as PLLA plasticizers, including glycerol [17], poly (ethylene glycol) [18], poly (propylene glycol) [19], acetyl triethyl citrate [20], glycerol monostearate [21], oligo (lactic acid) [22], β-cyclodextrin/d-Limonene [23], and *N*,*N*-diethyl-3-methylbenzamide (DEET) [24], with small molecules being more efficient than oligo- or polymers regarding the lowering of *T*_g_ of the host polymer. However, they may be unstable at the temperature used for melt processing, have a stronger tendency for phase separation from the host polymer, and tend to migrate toward surfaces during storage, thus leading to undesired changes in properties [25].

DEET, mentioned above, is an insect repellent [26,27], thus offering additional functionality beyond the basic plasticizing effect. The polymer/repellent system PLLA/DEET has been investigated in detail from the point of view of general phase behavior [28,29,30,31] as well as the generation of semi-finished or final products [24,32,33]. These investigations confirmed both a plasticizing effect of DEET on PLLA and, simultaneously, the possible use of PLLA as an insect repellent carrier. In the present work, we extend the initial work of modifying PLLA with insect repellents by employing ethyl butylacetylaminopropionate (IR3535) instead of DEET. Compared to the gold standard repellent DEET [26,27], IR3535 has fewer side effects on the environment and human beings, allowing possible use by pregnant women and children [34,35,36] to prevent mosquito-borne tropical diseases, such as malaria, which causes hundreds of thousands of deaths each year [37,38].

Recent studies have proven that PLLA can be used as a carrier/drug delivery reservoir for the repellent IR3535 [39,40,41]. Early investigations focused on the evaluation of the repellent/solvent-rich part of the PLLA/IR3535 system for the preparation of scaffolds for the controlled release of IR3535 into the environment [40]. IR3535 and PLLA form solutions at elevated temperature, and the solutions demix upon cooling via polymer crystallization caused by solid–liquid (S–L) thermally induced phase separation (TIPS) [42,43], leading to the formation of solid PLLA scaffolds. These scaffolds are tunable regarding pore size through both the crystallization temperature and the polymer content. Microporous scaffolds with different fine structures were obtained, which carried the mosquito repellent in intra- and interspherulitic pores. The intraspherulitic pore size of PLLA increased with crystallization temperature and decreased with the polymer content [40]. In the case of employing amorphous poly (d/l-lactic acid) (PDLLA), an in-depth analysis of the glass transition temperature by using fast scanning chip calorimetry combined with in situ evaporation of the liquid for controlled change in the system composition revealed that PDLLA and IR3535 are thermodynamically miscible over the entire composition range [39]. Crystallization of PLLA in the presence of IR3535 is faster than melt crystallization of neat PLLA, with the maximum crystallization rate increasing with the PLLA content over the investigated range of 5 to 50 mass % (m%) PLLA [40].

Important from the point of view of obtaining end user products, PLLA parts, such as finger rings accommodating up to 25 m% liquid mosquito repellent IR3535, were successfully fabricated by 3D printing, suggesting novel materials/processing routes to obtain PLLA-based controlled repellent release devices [44]. Furthermore, melt extrusion of PLLA/IR3535 strands as an efficient processing route for obtaining semi-finished products has been investigated, focusing on thermal, rheological, and release properties of the samples containing up to 25 m% IR3535 [41]. In the present work, we attempted to expand the initial melt extrusion study, so as to include—additional to the evaluation of the plasticizing effect of IR3535 and the release characteristics—the analysis of the long-term crystallization behavior and its effect on the mechanical performance.

## 2. Materials and Methods

### 2.1. Materials and Preparation

PLLA with less than 1% d-isomers, named L-175, was provided by Total-Corbion (Amsterdam, The Netherlands). The melt flow index of the material is reported as 8 g/10 min (210 °C/2.16 kg) [45]. IR3535, with a purity of 98%, was purchased from Carbolution Chemicals GmbH (St. Ingbert, Germany) [46] and used as received without further purification. It is a clear liquid at room temperature, with a glass transition temperature of close to −90 °C [39]. The estimated boiling points at atmospheric pressure and at 0.02 kPa are slightly below 300 °C and about 110 °C, respectively, and the vapor pressure is reported as being around 0.15 Pa at 20 °C [35,36].

PLLA was dried in a dry-air dryer at 60 °C for 4 h prior to processing. Strands of PLLA and IR3535 mixtures, as well as of neat PLLA, were prepared using a modular co-rotating and intermeshing twin screw extruder LTE20-44/00 (Labtech Engineering Co. Ltd., Samutprakarn, Thailand) with a screw diameter of 20 mm and an L/D ratio of 44. A special screw design with a low number of shear and kneading elements was used for processing. The throughput of the PLLA was variable, depending on the required repellent concentration in the compound (see Table 1). The dry PLLA was fed via the main hopper, and the screw speed for the preparation of the compounds was set at 250 1/min. The temperature profile of the extruder was set from the feeding zone (zone 1) to the die (zone 11) as follows: 185, 185, 185, 185, 185, 190, 190, 200, 200, 210, and 210 °C. The liquid repellent was added via a volumetric pump, preeflow^®^ eco-PEN600 (ViscoTEC Pumpen- u. Dosiertechnik GmbH, Töging am Inn, Germany), at the feeding zone. The melt was extruded through a dual-strand die with a diameter of 3 mm, before being cooled in a water bath at room temperature and subsequently pelletized. It is noted that thermal degradation of IR3535 under the extrusion conditions is excluded, based on dedicated IR3535 stability experiments described elsewhere [35].

### 2.2. Instrumentation

Thermogravimetric analysis (TGA): The content of the repellent and degradation behavior of PLLA in mixtures after extrusion, as well as the repellent release kinetics were investigated by a TGA 2 LF/1100/694 (Mettler Toledo, Greifensee, Switzerland). For non-isothermal repellent evaporation experiments, sample pieces with a mass of 7.0 ± 0.3 mg were prepared by cutting sections across the whole strand, that is, perpendicular to the extrusion direction, placing them into alumina crucibles, which were then heated from 30 to 600 °C at 5 K/min, using nitrogen as the purge gas, followed by heating from 600 to 900 °C at 30 K/min in an oxygen atmosphere. The gas flow rate, in both cases, was 50 mL/min. Measured data were automatically subtracted by a blank curve using the instrument software. For isothermal repellent release experiments, samples with a mass of 3.0 ± 0.2 mg were heated to predefined temperatures between 50 and 100 °C at a rate of 20 K/min in a nitrogen atmosphere, and then held at these temperatures for 24 h, to allow evaporation of the liquid repellent. Measured data were automatically compensated for buoyancy, using the instrument software.

Polarized light optical microscopy (POM): The structures of the PLLA/IR3535 mixtures after melt extrusion and after additional annealing at 60 °C for 3–4 weeks were observed using POM. Samples were cut into thin slices with a thickness of 20–30 µm using a CUT-5062 rotary microtome (Slee, Mainz, Germany). Then, thin sections were embedded in immersion oil between two cover glasses with a diameter of 15 mm and observed with a DMRX optical microscope (Leica, Wetzlar, Germany) in transmission mode using crossed polarizers. The images were captured with a Motic 2300 CCD camera attached to the microscope.

Scanning electron microscopy (SEM): A Vega 3 SBU SEM (Tescan, Dortmund, Germany) was employed for visualization of the structure of the PLLA/IR3535 mixtures after melt extrusion. Non-pelletized strands with a length of about 10 cm were placed in liquid nitrogen, kept there for 15 min, and subsequently cryo-fractured using a plier. The sample cross-sections were gold coated before measurements. The instrument was operated in the high vacuum mode, a tungsten cathode filament was used as an electron gun, and an acceleration voltage of 10 kV was applied.

Differential scanning calorimetry (DSC): DSC was employed to analyze the thermal behavior of PLLA/IR3535 mixtures after melt extrusion. Measurements were performed using a calibrated heat flux DSC 1 (Mettler-Toledo, Greifensee, Switzerland) equipped with an FRS5 sensor. The device was connected to a TC100 intracooler (Huber, Offenburg, Germany), to allow cooling at rates up to 30 K/min. The furnace was purged with nitrogen gas at a flow rate of 60 mL/min. Samples with a mass between 8 and 12 mg were placed into 40 μL aluminum pans with a pin and covered with a lid. Further information regarding thermal profiles is provided below.

Dynamic mechanical analysis (DMA): DMA was performed using an Anton Paar MCR 501 instrument equipped with a CDT600 oven and SRF5 clamps. For non-isothermal tests, rectangular bars with a dimension of 80 × 10 × 4 mm^3^ were tested at a frequency of 1 Hz in shear mode, using a 2 K/min temperature ramp from −40 to 140 °C. For measurements of the elastic modulus of extrudates at room temperature, the tensile mode was applied. The rectangular bars for DMA measurements were obtained from the extrudates after reprocessing by injection molding using an IM 12 micro-injector (Xplore, Sittard, Netherlands) in combination with a MC 15 twin-screw micro-compounder (Xplore, Sittard, Netherlands). The reprocessing melt temperature and number of revolutions of the screw were between 180 and 210 °C and 50 rpm, respectively. The mold/cavity temperature was room temperature. To assure that reprocessing did not cause evaporation of the repellent, its content was redetermined by TGA using the same test method as described above.

Tensile testing: Uniaxial tensile stress–strain tests according to the standard DIN EN ISO 527-2 [47] were performed on a Zwick/Roell Z020 instrument (ZwickRoell GmbH & Co. KG, Ulm, Germany) at 25 °C at a strain rate of 5 mm/min. The humidity during testing was 45.9%. Dumbbell-shaped specimens with a length of 40 mm, width of 5.0 mm, and thickness of 2 mm were used, and were obtained by reprocessing via injection molding, as described above. In this case, the repellent content in the mixtures after reprocessing was redetermined by TGA.

Fourier transform infrared (FTIR) spectroscopy: FTIR spectra were recorded on a Nicolet iN50 spectrometer/microscope (Thermo Scientific, Waltham, MA, USA) in reflection mode. Background-corrected spectra were measured at a resolution of 2 cm^−1^ in the wavenumber range of 4000 to 600 cm^−1^, averaging 64 scans. Thin sections of extrudates, cut perpendicular to the extrusion direction, with a thickness of 15 µm, were used for measurements, and the frame size of the microscope window was 300 × 300 µm^2^.

Small-angle and wide-angle X-ray scattering (SAXS and WAXS): X-ray scattering experiments were performed using a Retro-F laboratory setup (SAXSLAB, Massachusetts, USA), equipped with a microfocus X-ray source (AXO Dresden GmbH, Dresden, Germany) and ASTIX multilayer X-ray optics (AXO Dresden GmbH, Dresden, Germany) as the monochromator for CuKα radiation (λ = 0.154 nm), to obtain information about the presence of PLLA crystals and their polymorphic structure, formed after extrusion/additional annealing at 60 °C. The instrument was operated in transmission mode, and, as a sample holder, aluminum discs with a central hole of 2 mm in diameter were used. Measurements were recorded in vacuum using a PILATUS3 R 300 K detector (Dectris Ltd., Baden-Daettwil, Switzerland). The sample-to-detector distance was calibrated using silver behenate, and measurement times of 300 and 900 s were used for WAXS and SAXS analyses, respectively. The thickness of the cross-section of the extrudates used for the X-ray analyses was about 2 mm, and the beam diameter was 0.25 mm.

## 3. Results and Discussion

### 3.1. Repellent Content of Extrudates

The content of IR3535 in the various PLLA/IR3535 mixtures after melt extrusion was evaluated using TGA at a heating rate of 5 K/min and employing samples with an initial mass of around 7 mg. Note that the temperature of mass losses in TGA depends on the heating rate as well as the sample mass [41,44,48,49], with the rather low selected values minimizing thermal lag and increasing the resolution. The left and right plot of Figure 1 show the original TGA curves, percentage mass as a function of temperature, and their first derivative, respectively.

The first low-temperature mass loss event in all samples containing the repellent is due to the evaporation of the repellent IR3535. In the case of neat IR3535 (brown curve), mass loss starts at around 100 °C at the selected conditions, and evaporation is completed at around 230 °C. In PLLA/IR3535 mixtures, the mass loss due to evaporation of IR3535 is slightly delayed/shifted to higher temperature, due to entrapment of the repellent in the polymer matrix, affecting the vapor pressure and diffusion pathways [50,51,52]. This notwithstanding, when evaporation of IR3535 is completed, for all PLLA/IR3535 mixtures, a nearly constant mass plateau at around 300 °C is observed, allowing an estimation of the actual IR3535 content (see also the vertical arrow in the right plot of Figure 1, confirming zero slope of the TGA curves at around 300 °C). The table in the legend compares the expected IR3535 content, as anticipated with the amount of IR3535 added in the extrusion process (see Table 1), and the content of IR3535 measured by TGA. Considering minor errors in the determination of the effective IR3535 content in the mixtures due to the interplay of the kinetics of evaporation and the heating rate, the observed data suggest that the target and actual IR3535 concentrations in the extruded strands are very similar, except in the case of the sample with the highest amount on IR3535 (dark blue curve). It is assumed that the observed difference between 30 m% (target) and 23 m% (measured) is caused by minor evaporation of IR3535 during extrusion. In all other cases, obviously, distinct evaporation of the liquid repellent during melt mixing at the chosen extrusion parameters was absent. Further close inspection of the TGA curves reveals a two- or even threefold IR3535 evaporation event in the mixtures with PLLA, which is more easy to recognize in the derivative curves. This does not affect the determination of the actual IR3535 content.

The mass loss event at higher temperature in PLLA/IR3535 mixtures, at around 360 °C, being the minimum in the derivative curve, is caused by decomposition of PLLA, as concluded from the analysis of neat PLLA (black curve) and comparison with literature data [53,54]. The PLLA mass loss in all PLLA/IR3535 mixtures started at the same temperature, being independent of the composition, and suggests that the presence of the liquid repellent has a non-measurable effect on the thermal degradation kinetics of PLLA.

### 3.2. Morphology and Fracture Behavior of Extrudates

The morphology of neat PLLA and PLLA/IR3535 mixtures after melt extrusion was observed using POM and SEM. The left two images of Figure 2 are POM micrographs of PLLA/IR3535 extrudates containing 0 and 23 m% IR3535, with the images obtained from thin sections taken perpendicular to the extrusion direction of the strands with a diameter of about 2 mm. The micrographs are featureless and indicate the absence of polymer/liquid phase separation at the µm length scale. Though POM alone cannot prove/disprove the presence of crystals [55,56], at least the absence of spherulites, being a typical feature of PLLA melt crystallization at rather low and intermediate supercooling of the melt [57,58], is obvious.

The four images to the right in Figure 2 are SEM micrographs of cross-sectional surfaces of cryo-fractured extruded strands of neat PLLA (top) and of a PLLA/IR3535 mixture containing 23 m% IR3535 (bottom). The two left and right images provide an overview of the cross-section and of details of the structure at higher magnification, respectively. Obviously, the obtained surface of neat PLLA is rather smooth and flat, indicating brittle fracture, while the surface of the PLLA/IR3535 mixture containing 23 m% IR3535 seems structured, revealing reduced brittleness compared to unmodified PLLA. In addition, a skin–core morphology, which could develop during solidification of the strands in the presence of a temperature gradient when extruding into a water bath [59], as occasionally observed and possibly affecting the repellent release rate, is absent [60,61,62]. Additionally, in the case of the SEM images, polymer/liquid phase separation and polymer crystallization are not detected. It is worth noting that all samples listed in Table 1 were analyzed by POM and SEM, yielding qualitatively similar results. Summarizing these experiments, (i) macroscopic—at the µm length scale—polymer/repellent phase separation is absent, (ii) spherulitic superstructures related to polymer crystallization are not observed, and (iii) the presence of repellent reduces the tendency for brittle fracture behavior.

### 3.3. DSC Analysis of Extrudates

The left plot of Figure 3 presents DSC curves of neat PLLA and PLLA/IR3535 extrudates of different composition recorded at a heating rate of 20 K/min. The DSC curve obtained from neat PLLA displays, from low to high temperature, the glass transition at around 60 °C [63], overlapped by an enthalpy recovery peak due to physical aging below *T*_g_ [64,65,66], a cold crystallization peak at about 125 °C, and a melting peak at about 178 °C. Melt crystallization at 125 °C is typically connected with the formation of orthorhombic α-crystals [67,68,69,70], which then melt close to 180 °C, as expected for a PLLA grade with only a minor amount of d-isomers in the chain [71,72]. The total enthalpy change observed during heating, which equals the difference between the enthalpy of melting and the enthalpy of cold crystallization, is proportional to the crystal fraction in the sample at the beginning of the heating process in the DSC experiment, and is about zero; that is, the sample is fully amorphous after extrusion into the cold water bath. This observation fits the expectation derived from earlier analyses of the cooling rate dependence of PLLA crystallization [73,74,75].

The addition of IR3535 results in a gradual decrease in *T*_g_, which reaches 15.0 °C in the mixture containing 18 m% IR3535, indicating that IR3535 acts as a plasticizer for PLLA. In the case of the extrudate with 23 m% IR3535, the glass transition is not detectable, which may be related to the presence of crystals, reducing the heat capacity step at *T*_g_ (see top curve of the left plot of Figure 3). Note, red and blue color tones of DSC curves are used to indicate glass transition temperatures above and below room temperature, respectively; such color coding is applied whenever possible. In addition, the temperatures of cold crystallization and melting systematically decrease with increasing amounts of IR3535 in the mixtures, though the effect is low if the IR3535 content is less than 5 m%. The lowered PLLA crystallization and melting temperatures in the mixtures is probably caused by the equilibrium melting point depression, according to Flory [76,77,78,79], and the lowered glass transition temperature, shifting the temperature range of possible crystallization to lower values [80,81]. More important, from the point of view of obtaining information about the initial structure of the extrudates, is the observation that the area of the cold crystallization peak decreases with increasing IR3535 content in the mixtures, while the area of the melting peak remains unchanged. In other words, crystallization of PLLA at the given extrusion and 24-day room temperature storage conditions is enhanced in the presence of IR3535, presumably due to the lowered *T*_g_, widening the temperature range of possible crystallization and increasing the mobility of molecular segments. In the case of the sample that contains 23 m% IR3535, cold crystallization is completely absent, indicating that a high degree of crystallinity was already achieved before the DSC analysis. The right plot of Figure 3 shows similar results for injection-molded bars reprocessed from extrudates stored for around 4 months at room temperature, which were used for mechanical tests.

Quantitative data regarding the glass transition temperature and the PLLA-content-normalized enthalpy change during heating (enthalpy of melting + enthalpy of cold crystallization) are shown in the left and right plots of Figure 4, respectively, as a function of the PLLA content. The different colors/symbols represent data obtained from samples analyzed after storing/annealing at ambient temperature for about 1 month (black/gray squares) (see also the DSC scans of Figure 3), for 6 months (red/gray circles), and for 1 year (blue/gray triangles). In addition, green/gray downward triangles are data obtained from additionally reprocessed/injection-molded samples of neat PLLA and PLLA/IR3535 mixtures stored for 4 months at ambient temperature, which were used for tensile/DMA testing.

Regarding glass transition (left plot of Figure 4), it is observed that presence of IR3535 in mixtures with PLLA leads to a decrease in *T*_g_, pointing to miscibility at the length scale that is examined with the glass transition, that is, several nanometers [82,83]. Important in the context of later discussion of the crystallization behavior is the observation that the *T*_g_ of mixtures containing more than about 15–20 m% IR3535 is below room temperature (21 °C), which is indicated with the horizontal dashed gray line. For a more complete picture of the miscibility of PLLA and IR3535, the left plot of Figure 4 also contains data observed in earlier work (see diamond symbols), collected over the entire concentration range using a special experimental approach of successive evaporation of repellent and repeated *T*_g_ analysis, and additional *T*_g_ analyses of solvent-rich compositions, including neat IR3535 [39]. Though a different poly (lactic acid) (PLA) grade was used in that study—containing 50% d-isomers—the miscibility, obviously, is similar to the case of the highly stereoregular PLLA used here. Inspection of data for samples stored for different time periods before the DSC analyses reveals a negligible effect if the concentration of IR3535 is equal or lower than 9 m%. However, for the sample containing 18 m% IR3535, the glass transition became undetectable by DSC after long-term annealing at room temperature, which is likely caused by slow crystallization of PLLA. Since IR3535 cannot be included in the PLLA crystal phase, the amorphous phase probably enriches IR3535, shifting *T*_g_ further down to lower temperature (see arrow).

For samples containing less than 9 m% IR3535, the crystallinity is zero, and does not depend on the annealing time at room temperature. In contrast, for samples containing 9 m% IR3535, or more, a distinct effect of the storing time is seen, such that the crystallinity increases with time (right plot of Figure 4, see black arrow). Furthermore, the data allow to conclude that crystallization proceeds faster if the repellent content is higher. While the crystallinity-increase within 1 year of storing the extrudate at ambient temperature is marginal for the sample containing 9 m% IR3535, in the case of the sample containing 23 m% repellent, the maximum possible crystallinity, as indicated with the plateau, is already achieved after short-term annealing for about 1 month. With the knowledge of the bulk enthalpy of melting of PLLA α-crystals [84], and its temperature dependence [85], of close to 100 J/g for crystals melting slightly below 180 °C (see Figure 3), a PLLA crystallinity of almost 70% can be achieved.

### 3.4. Thermal Stability of Extrudates

The thermal stability of IR3535 after extrusion was confirmed by FTIR measurements, as presented with the left plot in Figure 5. For neat IR3535, the strongest bands of the spectra are located at 1736 cm^−1^ and 1651 cm^−1^ assigned to the carbonyl stretch vibrations of the ester and amide groups, respectively. The band associated to the IR3535 amide group is well-separated from the PLLA bands, allowing identification of IR3535 in the spectra obtained from PLLA/IR3535 extrudates. As expected, in the PLLA/IR3535 mixtures, the intensities of these bands increase with the IR3535 content. More important, when comparing the spectra of extruded mixtures with the spectrum obtained from neat IR3535, there is no shift of the characteristic bands observed, indicating that the repellent was able to withstand the thermomechanical history during processing. Independent studies of the thermal stability of IR3535, available in the literature [35,44], in contrast, suggested slight oxidative degradation by observation of a new band near 1690 cm^−1^ when exposed to air for four months at 50 °C. Here, in our study, new carbonyl bands are not observed.

The right plot of Figure 5 shows details of the FTIR spectra in the wavenumber range between 890 and 970 cm^−1^, which allows one to obtain information about the possible presence of ordered PLLA structures. As such, PLLA/IR3535 extrudates with 82 and 77 m% PLLA show an additional band at 922 cm^−1^ (see dashed line) that is assigned to PLLA crystals (see right plot) [86,87,88,89]. This band is not detected for other extrudates of lower repellent content, and indicates the absence of crystals after storing these samples at room temperature for 6 months. These results are consistent with DSC data shown above.

### 3.5. WAXS and SAXS Analysis

X-ray scattering was used to assess the structure of extrudates, including the crystallinity and crystal structure of PLLA. The left plot of Figure 6 shows WAXS curves of extrudates of PLLA containing different amounts of IR3535, stored at room temperature for about 9 months (bold curves), and of samples stored at room temperature for 17 days and additionally annealed for 3–4 weeks at 60 °C after extrusion (thin curves; labelled ‘x-ann’, with x representing the PLLA content). The initial purpose of annealing the extrudates at 60 °C was to remove the repellent, followed by observation of the morphology in order to trace the location of the repellent. However, by chance, it was found that the samples changed their appearance from clear to turbid, except in the case of neat PLLA and the extrudate with 77 m% PLLA. With the suspicion that turbidity was caused by crystallization, these samples were also investigated by X-ray scattering. Note that annealing at 60 °C was performed until the mass of the samples was constant.

The WAXS patterns of non-annealed, neat PLLA and extrudates containing less than 18 m% IR3535 display a broad amorphous halo, which confirms the lack of crystals, as was also concluded from the DSC analysis (see Figure 4, right). However, for samples annealed at 60 °C, distinct scattering peaks are observed, except for neat PLLA, which only shows a single small peak (see arrow).

Long-term annealing of the extrudates at 60 °C shows a major effect on the structure of the samples containing 5 and 9 m% IR3535 (red and light red curves, respectively). In these two cases, non-annealed samples displayed only an amorphous halo, while after annealing intense scattering peaks are detected, proving the formation of crystals during the annealing step.

Inspection of the sets of peaks detected, we assume that α-crystals are the predominant crystal form. This assumption is mainly based on scattering peaks measured at scattering angles slightly higher than 12 and 22 deg 2θ [69,90,91]. This observation was not expected since in the case of neat PLLA low-temperature crystallization typically yields disordered α’-crystals. Obviously, the presence of the dissolved liquid repellent supports the formation of the more ordered α-crystal form; similar results have been observed for solution-crystallized PLLA, regardless the crystallization temperature [28,40,92]. These results are consistent with DSC heating curves (not shown) obtained from annealed samples, which did not display the typical exothermic α’- to α-crystal transformation peak [93], except in the case of the sample containing 2 m% IR3535.

Regarding the PLLA/IR3535 extrudates with 82 and 77 m% PLLA, the cylindrical samples with a diameter of about 2 mm and a length of 2 mm were measured in two directions, with the beam parallel (p) and vertically (v) oriented with respect to the extrusion direction, marked ‘p’ and ‘v’ in the 2D WAXS scattering patterns (right plot of Figure 6), respectively. The results indicate that crystals in the PLLA/IR3535 extrudates do not show preferred orientation.

The left plot of Figure 7 shows Lorentz-corrected SAXS curves of PLLA/IR3535 extrudates stored at room temperature for about 9 months (bold curves) and after additional annealing at 60 °C for 3–4 weeks (thin curves), respectively. The curves were evaluated by calculating the interface distribution function [94,95], revealing information about the long period (*LP*), and the thickness of lamellae (*d*_c_) and of the amorphous layer (*d*_a_), presented as a function of the sample composition with the right plot in Figure 7. Except for neat PLLA, all curves in the left plot show a distinct long-period maximum, which corresponds to a distance of about 16 nm, and indicating formation of stacks of lamellae. It appears that the presence of IR3535 enhances the formation of lamellar stacks since the intensity of the long-period maximum increases with its content; in the case of the cold-crystallized, annealed samples containing 9 and 18 m% IR3535, even higher-order long-period maxima are observed. Inspection of the position of the long-period maximum reveals an only minor effect of the sample composition: *LP* decreases from 17 nm in PLLA containing 5 m% IR3535 to around 15 nm in the case of the sample containing 23 m% IR3535. Similarly, the thickness of lamellae decreases from around 14 nm to 11 nm, respectively, while the amorphous layer thickness is only 3 nm, being almost independent of the IR3535 content. Calculation of the linear crystallinity (*d*_c_/(*d*_a_ + *d*_c_) × 100%) [96,97], then yields values of around 80% for all samples containing between 5 and 18 m% IR3535. This value confirms a recent study about the SAXS crystallinity of PLLA [84].

### 3.6. POM Analysis of Annealed PLLA/IR3535 Extrudates

Figure 8 shows POM images of PLLA/IR3535 extrudates with a maximum content of IR3535 of 9 m%, annealed at 60 °C for 3–4 weeks. Preparation of thin sections of samples containing higher amounts of IR3535 was complicated due to their low glass transition temperature (see Figure 4, left), being below RT; the images are therefore not shown. Upper and lower images in Figure 8 were observed at different magnification, as indicated with the scale bars; the upper images, in fact, provide an overview of the cross-section of the pellets. WAXS data of neat PLLA suggested a fully amorphous state, and as such the images are featureless and black when observed with the sample located between crossed polarizers. In the case of samples containing IR3535, crystallization occurred, and the POM micrographs show numerous white spots due to birefringence related to the presence of crystals. Such morphology is probably related to the nucleation pathway [55,56,98,99]. Fast cooling, followed by long-term annealing near *T*_g_, causes the generation of a large number of homogeneous nuclei, which then, upon reheating, grow to crystals, with similar morphologies also reported for PLLA in the literature [73,89]. Due to the high nuclei number, growth of large superstructures, such as spherulites, is then not possible, causing the observed spotty structure. A distinct and systematic effect of the repellent concentration, however, is not observed. Note that we do not assume that the white spots are single crystals, rather that we consider them as aggregates of lamellae/lamellar stacks.

### 3.7. Mechanical Properties of PLLA/IR3535 Extrudates

In order to finalize the material for an engineering product or industrial application, mechanical properties are a very important consideration. Figure 9 shows the elastic modulus of injection-molded bars, reprocessed from extrudates, at room temperature evaluated by tensile stress–strain testing and DMA in tensile mode, respectively. The data of both tests are consistent and reveal a constant elastic modulus for PLLA/IR3535 extrudates containing between 2 and 9 m% IR3535, a strong decrease in the IR3535 concentration range of 9 to 18 m%, and then constancy when increasing the IR3535 content to 23 m%. The obtained modulus of neat PLLA and for mixtures with low amounts of IR3535 agrees with values available in the literature, being slightly higher than 3 GPa [100,101], indicative of presence of a glassy amorphous phase. The dropdown of the modulus of elasticity at repellent concentrations higher than 5 m% is caused by the change in *T*_g_ to near or even below the measurement temperature of around 22 °C (see also Figure 4, left) and the transition of the PLLA glass into the rubbery state, regardless the presence of crystals.

Figure 10 depicts the loss factor tan-δ (left plot) and shear modulus (right plot) as a function of temperature, respectively, estimated by DMA using a 2 K/min temperature ramp from −40 °C to 140 °C. In the left plot of Figure 10, the peak temperature in the tan-δ curves is regarded as *T*_g_, confirming the trend obtained by DSC analyses described above. For neat PLLA, *T*_g_ is around 65 °C, and then it decreases with increasing IR3535 content, first slightly for repellent concentrations up to 9 m%, and then stronger for higher IR3535 concentrations; for the samples containing 18 and 23 m% IR3535, tan-δ maxima are hardly observed.

As presented in the right plot of the Figure 10, at low temperatures, PLLA/IR3535 injection-molded bars, reprocessed from extrudates, with a PLLA content between 91 and 100 m% display a shear modulus *G*´ of about 1.3 GPa (around 1/3 of the tensile modulus presented in the Figure 9) at low temperatures, with slight variations among the different samples. By increasing the temperature, a sudden drop of *G*´, associated with the glass transition of the amorphous phase of PLLA containing IR3535, is detected, with the temperature of the dropdown decreasing with increasing IR3535 content.

Upon further increase in the temperature, at about 80 °C for neat PLLA and lower temperatures for samples with less than 23 m% IR3535, PLLA chains gain sufficient mobility to allow cold crystallization, resulting in an increase in *G*´, being in general accord with the DSC observations (see the right plot of Figure 3). The minor difference between the temperatures of (kinetically controlled) cold crystallization obtained by DSC and DMA is caused by the different heating rates in these experiments [102,103]. After completion of cold crystallization, the modulus scales with the degree of crystallinity achieved in the various samples, being highest in the case of neat PLLA, since the crystalline phase is much stiffer than the rubbery amorphous phase [104,105,106]. In terms of the extrudates with 82 and 77 m% PLLA, the shear modulus at all temperatures is much lower than in the other samples, and cold crystallization is less distinct or even completely absent since crystallization almost completed before the DMA experiment (see the right plot of Figure 3).

Summarizing the DMA and tensile tests, liquid IR3535 acts as a plasticizer for PLLA, causing a decrease in *T*_g_ and affecting mechanical properties, in particular at temperatures slightly above room temperature. At higher repellent concentration of around 20 m%, a rather strong effect was also observed at ambient temperature (see vertical gray line in the right plot of Figure 10), making the material soft and ductile/non-brittle.

### 3.8. Repellent Release

For application of PLLA/IR3535 extrudates as repellent release devices, as well as plasticized/ductile material, knowledge regarding the release rate/evaporation characteristics at ambient and body temperature is of major importance. The kinetics of the repellent release from the polymer matrix was investigated by isothermal TGA experiments at temperatures ranging from 50 °C to 100 °C, allowing extrapolation of the release behavior of repellent at room temperature/body temperature. This temperature range is selected because the repellent release rate is accelerated by increasing the temperature while the sample structure is not damaged/changed.

Figure 11 presents the repellent release of PLLA/IR3535 mixtures initially containing 18 m% IR3535, stored at room temperature for about 10 months, at different release temperatures between 50 and 100 °C. This sample was selected since, on one hand, absence of bleeding before analysis was assured, while on the other hand, the repellent content appears sufficient from the point of view of using it as a repellent delivery device. In terms of the temperature dependence of the repellent release, the IR3535 release rate increases with increasing evaporation temperature. At an evaporation temperature of 100 °C (bottom curve), there is still some IR3535 left even after 24 h. However, at relatively low temperatures of, e.g., 50 °C, only about 14% out of the total initial repellent content evaporated within the analyzed time frame of 24 h, which suggests that the repellent release at temperatures below 50 °C may last many days, and thus, is difficult to measure by TGA. For this reason, we attempted to obtain the experimentally accessible temperature dependence of characteristic release times τ between 50 and 100 °C, and then extrapolated to the temperature of interest. Release time constants τ were determined by fitting the experimental mass loss curves with a single-exponential decay function, allowing their extrapolation and observation of a characteristic release constant τ, in this case the time of reducing the IR3535 mass to 1/e × 100% of its initial value, that is, the time when sample released 63.2% ((1 − 1/e) × 100%) of the initial IR3535 content. However, it seems that this approach is not applicable for release experiments performed between 80 and 100 °C, as the obtained TGA curves exhibit a double exponential decrease in the sample mass as a function of time, with a fast initial process followed by a much slower release. This may be caused by a change in the physical structure of the polymer during heating, such as cold crystallization or crystal reorganization, or by a change in intermolecular interactions between the polymer and repellent. As such, a single exponential decay function was used to fit the data obtained at temperatures between 50 and 70 °C [107], while for the experiment performed at 100 °C, the time at which 63.2% of the initial repellent evaporated was directly read from the measured TGA curve. For measurements performed at 80 and 90 °C, even after 24 h, the release was less than 63.2% of the initial repellent content; therefore, the release time constants were not determined.

The left plot of Figure 12 shows characteristic repellent release time constants τ of the PLLA/IR3535 extrudate initially containing 18 m% IR3535 as a function of the release temperature. The data suggest a non-linear increase in the characteristic evaporation time with decreasing temperature, being few hours at 100 °C and more than 100 days at 50 °C. Apparently, there is observed an exponential temperature dependence of time constants, pointing to an Arrhenius-type change in diffusion rate constants with temperature. Quantitative information regarding the release kinetics is obtained by plotting the logarithm of the inverse characteristic time, log (1/τ), as a function of the inverse release temperature, 1/*T*, as shown in the right plot of Figure 12. There is a linear dependence of data observed, suggesting characteristic repellent release times of 1.4 and 5.0 years at body temperature (37 °C) and room temperature (21 °C).

## 4. Conclusions

The present work is a continuation of our research efforts to contribute to the development of wearable insect repellent delivery devices, by incorporating liquid insect repellents into biosourced and biodegradable polymers, for slow subsequent release into the environment. With the focus on the polymer/repellent system composed of poly (l-lactic acid) (PLLA) and ethyl butylacetylaminopropionate (IR3535), PLLA/IR3535 mixtures at various compositions were prepared via melt extrusion technology, with the advantage of cost-efficient large-scale processing.

In the analyzed composition range of 2 to 23 m% IR3535, PLLA and IR3535 are miscible. Liquid IR3535 acts as a plasticizer for PLLA, causing a decrease in *T*_g_ and affecting the morphology, crystallization, and mechanical properties of the host polymer. PLLA/IR3535 extrudates containing between 2 and 9 m% IR3535 are amorphous, even after storing at room temperature for around 1 year after extrusion, while PLLA/IR3535 extrudates with 18 and 23 m% IR3535 showed an increase in the crystallinity over time, due to the low glass transition temperature. Despite the increase in the crystallinity, however, the presence of IR3535 softens PLLA and decreases its elastic modulus.

Furthermore, with the slow release of the repellent to the environment over a period of months or years, it is expected that the initial decrease in the glass transition temperature reverts. Both repellent-induced crystallization and evaporation of the repellent are expected to cause embrittlement, thus being disadvantageous regarding mechanical properties. On the other hand, the slow release of the repellent offers the opportunity to use extrudates as delivery devices, if engineered such that the mechanical performance is not important.

## Figures and Tables

**Figure 1 pharmaceutics-14-02381-f001:**
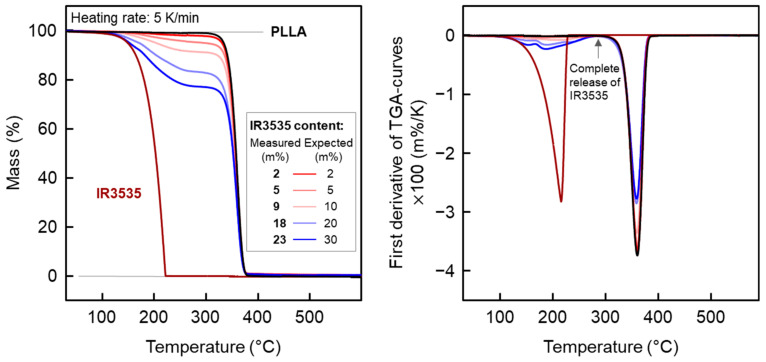
TGA heating curves normalized by the initial mass (**left**) and first derivative of the TGA curves (**right**) of IR3535 and PLLA/IR3535 extrudates, upon heating at 5 K/min in a nitrogen atmosphere. The legend in the left plot provides the measured (**left column**) and expected (**right column**) mass percentage of IR3535 in the extrudates, as derived from the plateau value after evaporation of IR3535. Measurements were performed after storing the strands at room temperature for about 32 weeks after extrusion.

**Figure 2 pharmaceutics-14-02381-f002:**
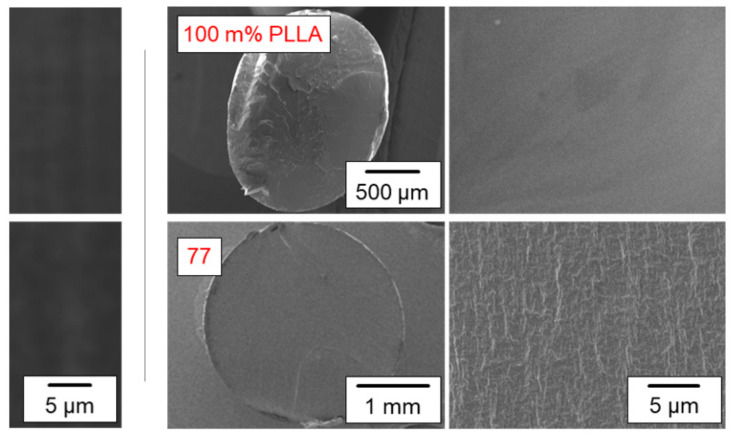
Morphology of extruded strands of neat PLLA (**top** images) and of a PLLA/IR3535 mixture containing 23 m% IR3535 (**bottom**), as observed by POM (**left**) and SEM (**right**), respectively, at different magnifications. The POM images were obtained from the central part of thin sections taken perpendicular to the extrusion direction of the strands with a diameter of about 2 mm, and the SEM images were obtained from cross-sections of cryo-fractured strands coated by gold, providing an overview and details of the structure. The PLLA/IR3535 extrudates for POM and SEM measurements were stored at room temperature for 3 weeks and 7–8 weeks, respectively.

**Figure 3 pharmaceutics-14-02381-f003:**
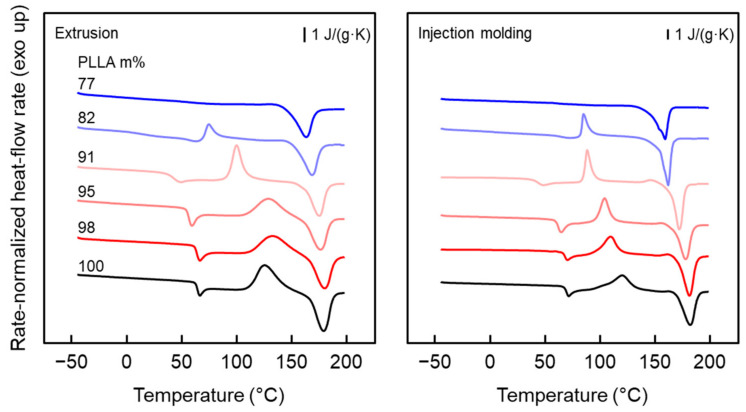
DSC curves: Rate-normalized heat flow rate as a function of temperature for extrudates of neat PLLA and PLLA/IR3535 mixtures (**left**) and injection-molded bars reprocessed from extrudates (**right**) heating at a rate of 20 K/min. Before measurement, samples were stored for 24 days (**left**) and around 4 months (**right**) at room temperature. Note, red and blue color tones are used to indicate glass transition temperatures above and below room temperature, respectively.

**Figure 4 pharmaceutics-14-02381-f004:**
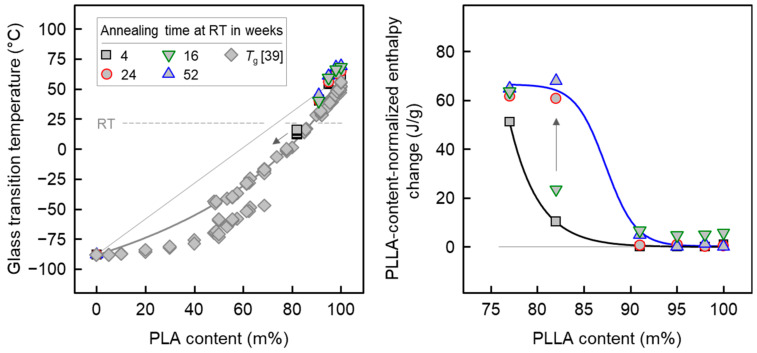
Glass transition temperature (**left**) and change in the enthalpy, normalized to the polymer content, during heating (**right**) of PLLA/IR3535 extrudates as a function of the polymer concentration, as derived from DSC data, exemplarily shown with Figure 3. The different colors/symbols represent data obtained from samples after storing at room temperature for different time, as indicated in the legend. For comparison, the left plot contains additional *T*_g_-data obtained from a system composed of non-crystallizable poly (lactic acid) (PLA) and IR3535, including a fit using the Gordon-Taylor equation, adapted from Polymer, 209, Fanfan Du, Christoph Schick, René Androsch, Full-composition-range glass transition behavior of the polymer/solvent system poly (lactic acid)/ethyl butylacetylaminopropionate (PLA/IR3535^®^), 123058, Copyright (2020), with permission from Elsevier [39]. Green/gray downward triangles are data obtained from reprocessed (by injection molding) and subsequently for 4 months stored samples of neat PLLA and PLLA/IR3535 mixtures, later on used for tensile/DMA testing.

**Figure 5 pharmaceutics-14-02381-f005:**
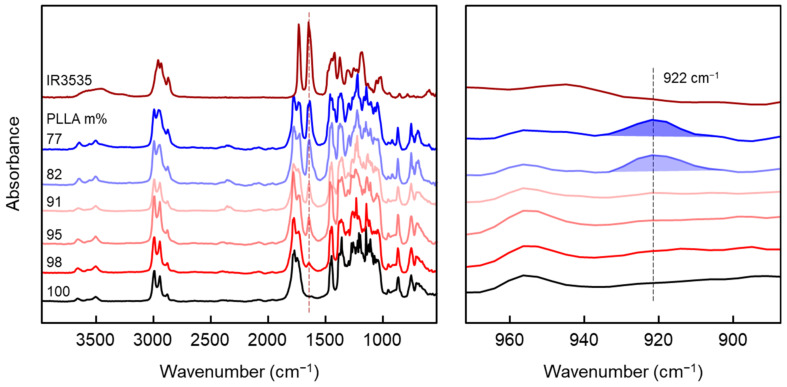
FTIR spectra of PLLA/IR3535 extrudates stored at room temperature for 6 months (**left**). The **right** plot shows details in the wavenumber range of 890 to 970 cm^−1^.

**Figure 6 pharmaceutics-14-02381-f006:**
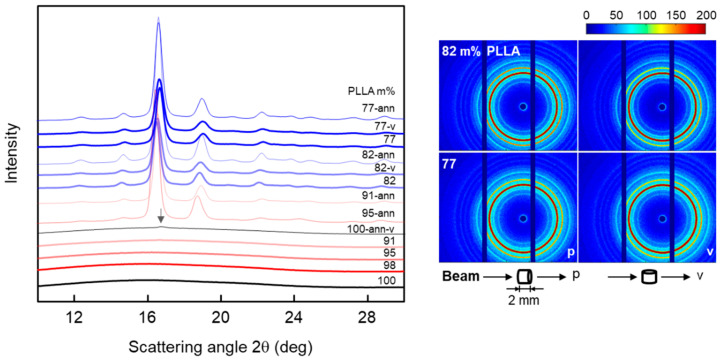
1D (**left**) and 2D WAXS patterns (**right**) of PLLA/IR3535 extrudates stored at room temperature for about 9 months (bold curves), and 1D WAXS patterns of extrudates stored at room temperature for 17 days and additionally annealed at 60 °C for 3–4 weeks (thin curves), with the latter labeled ‘x-ann’ and with x indicating the polymer content in mixture. The cylindrical samples with a diameter of about 2 mm and a length of 2 mm were measured in two directions, with the beam parallel (p) and vertical (v) to the extrusion direction, marked ‘p’ and ‘v’ in the 2D scattering pattern (**right**).

**Figure 7 pharmaceutics-14-02381-f007:**
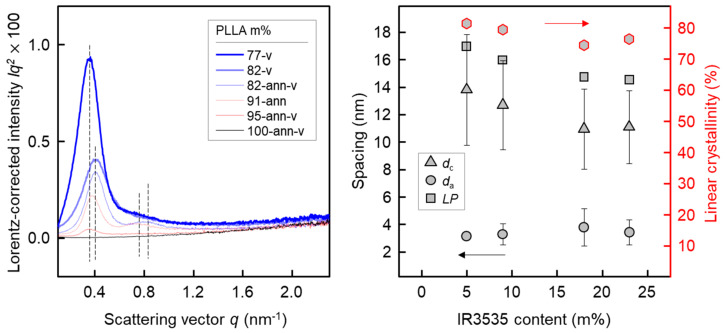
Lorentz-corrected SAXS curves of PLLA/IR3535 extrudates of different composition as indicated in the legend, stored at room temperature for about 9 months and annealed at 60 °C for 3–4 weeks, respectively (**left**). Long period (*LP*), and thickness of lamellae (*d*_c_) and amorphous regions in lamellar stacks (*d*_a_) as a function of the concentration of IR3535 in annealed PLLA/IR3535 extrudates (**right**). The red symbols represent the linear crystallinity (right axis). The bars represent the distribution of *d*_c_ and *d*_a_.

**Figure 8 pharmaceutics-14-02381-f008:**
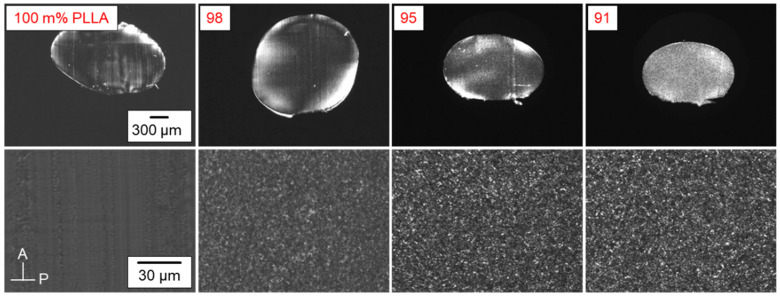
POM images of PLLA/IR3535 extrudates stored at room temperature for 17 days and then annealed at 60 °C for 3–4 weeks.

**Figure 9 pharmaceutics-14-02381-f009:**
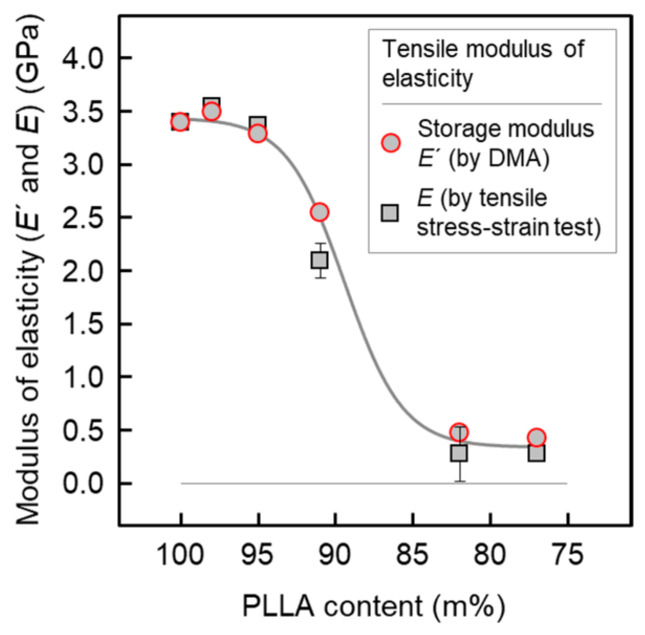
Modulus of elasticity of PLLA/IR3535 of injection-molded bars, reprocessed from extrudates, and measured by tensile stress–strain testing (black squares) and DMA (red circles) at room temperature. The line is drawn to guide the eye.

**Figure 10 pharmaceutics-14-02381-f010:**
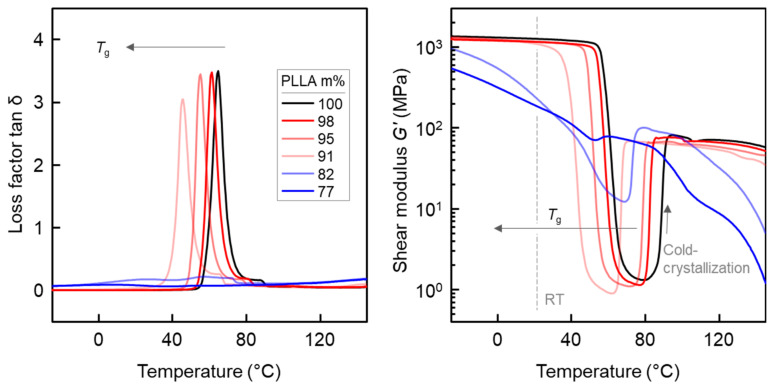
DMA curves, loss factor tan-δ (**left**) and shear modulus *G*’ (**right**) as a function of temperature, respectively, of PLLA/IR3535 extrudates upon heating at 2 K/min.

**Figure 11 pharmaceutics-14-02381-f011:**
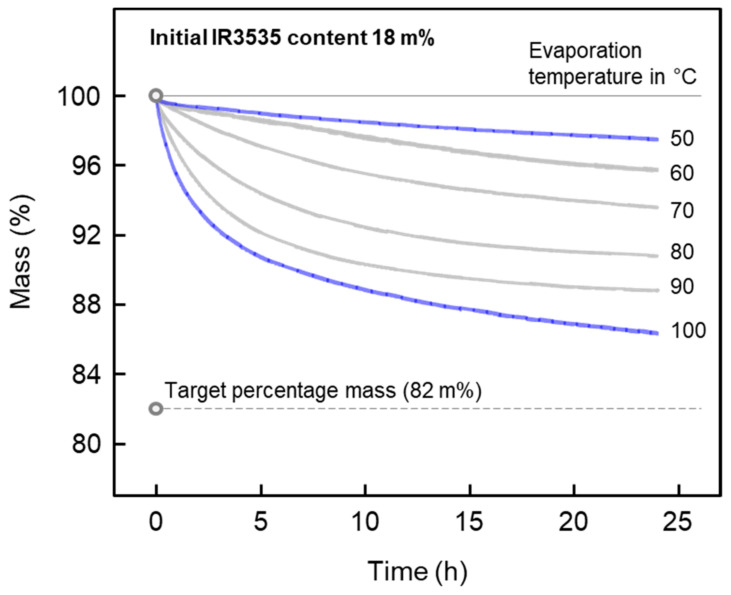
Percentage mass of a PLLA/IR3535 extrudate initially containing 18 m% IR3535 as a function of time during annealing at temperatures between 50 and 100 °C.

**Figure 12 pharmaceutics-14-02381-f012:**
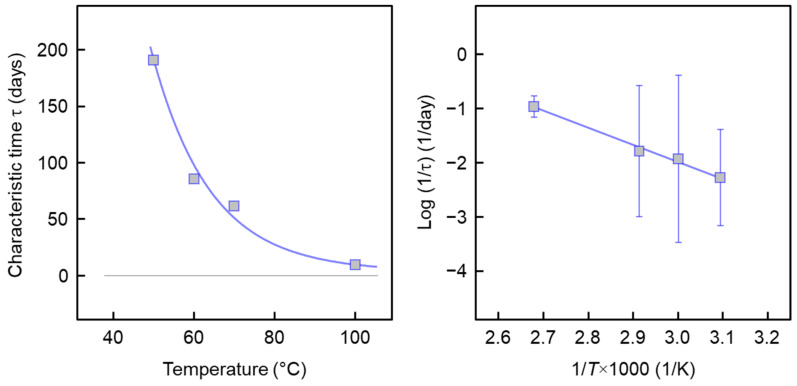
Characteristic repellent release time of strands with a diameter of about 2 mm of PLLA/IR3535 extrudates initially containing 18 m% IR3535 as a function of temperature (**left**), and logarithm of the inverse of the characteristic time, log (1/τ), as a function of inverse release temperature (1/*T*) (**right**). Each data point represents the average of two measurements; the error bars are smaller than the size of the symbols, and are therefore not shown.

**Table 1 pharmaceutics-14-02381-t001:** Sample composition and extrusion parameters.

PLLA/IR3535 Ratio (m%)	Screw Speed (1/min)	Main Feeder (1/min)	Engine Load (%)	Die Pressure (bar)	Melt Temperature (°C)				
					Die	Zone 2	Zone 4	Zone 6	Zone 8
100/0	250	17.0	61	10	190	186	187	191	199
98/2	250	35.8	88	21	189	186	184	191	199
95/5	250	20.8	58	12	190	184	187	190	199
90/10	250	9.8	38	5	186	182	185	189	198
80/20	250	13.1	36	5	188	182	184	187	197
70/30 ^1^	250	7.6	10	1	187	182	184	187	197

^1^ The surface of the sample was wet when storing at room temperature.

## Data Availability

Data will be made available on request.

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
