# Peer review of "Structure, Properties, and Release Kinetics of the Polymer/Insect Repellent System Poly (l-Lactic Acid)/Ethyl Butylacetylaminopropionate (PLLA/IR3535)"

_pharmaceutics, 2022, doi:10.3390/pharmaceutics14112381_

Round 1

Reviewer 1 Report

This paper reports a thermogravimetric analysis of repellent-release kinetics in Poly(L-lactic acid)/Ethyl butylacetyla-3 minopropionate (PLLA/IR3535) solid mixtures. In combination with other experimental measurements such as POM, SEM, SAXS, WAXD, DSC, FTIR and mechanical properties, the suitability of this melt-extruded material for wearable insect-repellent-delivery devices has been carefully evaluated. The results show optimistic for its practical usage. The work is interesting and worthy of publishing. The paper has been well prepared. So this reviewer suggests to accepting this paper with minor revision as suggested below.

1. Figures 1 and 10 may consider the visibility under black/white printing;

2. Several plots may not allow the time axes below zero, or the fraction axes beyond 100%.

Reviewer 2 Report

The work performed presents results related to the changes in the PLLA/IR3535 repellent delivery system caused by storage time. The knowledge of how time affects systems is an important issue in the production and commercialisation of delivery systems. This system has already been studied by the authors with results already published but not in the context of long-term behaviour. The work presents a detailed characterization of the systems. However some points must be cleared, before publication:

 -  The manuscript has excessive references in general and self-citations in particular.

- Several characterization methods were used to characterize the obtained systems after storage at room temperature or after annealing at 60 °C for 3–4 weeks. Do the authors have data for samples immediately after they are produced (DSC, FTIR, SAXS)? it would be interesting to compare with the results obtained after storage, even if only for some of the compositions, so as not to overload the graphics.

-  Why are some characterization results obtained after24 days, others after 4 or 32 weeks or after 6 months ? And the release behaviour after 10 months? It would be better to have the characterization results in all the technique used for the same storage or annealing period

- Figure 2 shows the morphology of extruded strands observed by POM and SEM. It is not clear in what conditions (before storage? after storage?) were the images obtained. In section 2.2 is mentioned that the structure of PLLA/IR3535 mixtures obtained by POM were observed after melt-extrusion and after additional annealing at 60 °C for 3–4 weeks. Please explicit this in the legend as in legend of Figure 8.

- Please remove reference 103 since it is not yet published.

- In pag 13, line 530, is stated “… presented in Figure 10 [104].” Correct to Figure 9. I suppose that data in Figure 9 were obtained in this work, so why is the reference 10 cited?

- In page 14, please correct the sentences “since crystallization almost completed before begin of the DMA experiment” (line 544-545) and “…there is observed a rather strong effect also at ambient temperature (see vertical grey line in the right plot of Figure 10) such that the material becomes soft and ductile/non-brittle.”  (line 550-552)

- Conclusions: Several conclusions presented are conclusions of previous work, already published, such as:

“…With the focus on the polymer/repellent system composed of PLLA and IR3535, an attempt is made to employ the melt-extrusion technology as an alternative to 3D-printing, fiber-spinning, and electro-spinning, with the advantage of cost-efficient large-scale processing.”

However, the melt-extrusion technology was already applied in [41] as mentioned by the authors in pag 2, lines 94-97

“We showed that it is possible to achieve repellent loadings up to 20–30 m% in the polymeric carrier, which, based on independent studies, is expected sufficient to yield repellency.” In [41] loadings of up to 25 %m were obtained.

“The repellent IR3535 dissolves in the amorphous PLLA phase and leads to a distinct decrease of the glass transition temperature, thus acting as a classical plasticizer.” This was already concluded in previous work already published.

 Concerning the conclusion “On the other side, the slow release of the repellent offers the opportunity to use extrudates as delivery-devices…”.  Is the fact of expecting a release period of months to years compatible with an efficient use as a repellent on the human body? With such a slow evaporation-release rate, will IR3535 reach concentrations that are perceptible to mosquitoes for the repellency to be effective?

Round 2

Reviewer 2 Report

After the authors answers and corrections performed I accept the manuscript in the present form.